# Spatial Distribution of Private Gene Mutations in Clear Cell Renal Cell Carcinoma

**DOI:** 10.3390/cancers13092163

**Published:** 2021-04-30

**Authors:** Ariane L. Moore, Aashil A. Batavia, Jack Kuipers, Jochen Singer, Elodie Burcklen, Peter Schraml, Christian Beisel, Holger Moch, Niko Beerenwinkel

**Affiliations:** 1Department of Biosystems Science and Engineering, ETH Zurich, Mattenstrasse 26, 4058 Basel, Switzerland; ariane.moore@bsse.ethz.ch (A.L.M.); aashil.batavia@usz.ch (A.A.B.); jack.kuipers@bsse.ethz.ch (J.K.); jochen.singer@bsse.ethz.ch (J.S.); elodie.burcklen@bsse.ethz.ch (E.B.); christian.beisel@bsse.ethz.ch (C.B.); 2SIB Swiss Institute of Bioinformatics, Mattenstrasse 26, 4058 Basel, Switzerland; 3Department of Pathology and Molecular Pathology, University of Zurich and University Hospital Zurich, Schmelzbergstrasse 12, 8091 Zurich, Switzerland; Peter.Schraml@usz.ch

**Keywords:** intra-tumour heterogeneity, private mutations, clonal exclusivity

## Abstract

**Simple Summary:**

Tumours consist of multiple groups of similar cells resulting from differing evolutionary trajectories, i.e., subclones. These subclones are prevalent in clear cell renal cell carcinoma (ccRCC). The aim of this study is to determine how similar or dissimilar the subclones in 89 ccRCC tumours are from one another regarding their gene mutations and expression profiles, i.e., the extent of intra-tumour heterogeneity. The implications of these alterations with respect to signalling pathways is also assessed. Deep sequencing allows for the identification of mutations with low-allele frequencies, providing a more comprehensive view of the heterogeneity present in the tumours. With an average of 62% of mutations having been identified in only one of the two biopsies, some of which in turn are found to impact gene expression, the complex makeup of ccRCC tumours is evident, and this can drastically influence treatment outcome.

**Abstract:**

Intra-tumour heterogeneity is the molecular hallmark of renal cancer, and the molecular tumour composition determines the treatment outcome of renal cancer patients. In renal cancer tumourigenesis, in general, different tumour clones evolve over time. We analysed intra-tumour heterogeneity and subclonal mutation patterns in 178 tumour samples obtained from 89 clear cell renal cell carcinoma patients. In an initial discovery phase, whole-exome and transcriptome sequencing data from paired tumour biopsies from 16 ccRCC patients were used to design a gene panel for follow-up analysis. In this second phase, 826 selected genes were targeted at deep coverage in an extended cohort of 89 patients for a detailed analysis of tumour heterogeneity. On average, we found 22 mutations per patient. Pairwise comparison of the two biopsies from the same tumour revealed that on average, 62% of the mutations in a patient were detected in one of the two samples. In addition to commonly mutated genes (*VHL*, *PBRM1*, *SETD2* and *BAP1),* frequent subclonal mutations with low variant allele frequency (<10%) were observed in *TP53* and in mucin coding genes *MUC6*, *MUC16*, and *MUC3A*. Of the 89 ccRCC tumours, 87 (~98%) harboured private mutations, occurring in only one of the paired tumour samples. Clonally exclusive pathway pairs were identified using the WES data set from 16 ccRCC patients. Our findings imply that shared and private mutations significantly contribute to the complexity of differential gene expression and pathway interaction and might explain the clonal evolution of different molecular renal cancer subgroups. Multi-regional sequencing is central for the identification of subclones within ccRCC.

## 1. Introduction

Tumours consist of genetically and phenotypically distinct cancer cell populations that evolve over time through a process that involves mutation and selection [1]. The presence of intra-tumour heterogeneity is well founded in renal cell carcinoma (RCC) with multiple subclones in both the primary tumour and paired metastasis [2,3,4,5,6]. Gerlinger et al. assessed the heterogeneity within 10 renal carcinomas, applying multi-regional sequencing. A large degree of intra-tumour heterogeneity with respect to both somatic mutations and somatic copy number variations was observed in all 10 tumours, with 75% of driver events found to be subclonal [4,7]. Martinez et al. further showed in eight RCC that the diversity within tumours is in some cases as high as the diversity between patients [8]. Therefore, the number of somatic mutations may be undervalued when taking a single biopsy from a solid tumour with only a subset of clones being present in the metastasis. This work laid the foundation for the TracerX consortiums analysis of 101 RCCs with 1206 multi-regional samples. When assessing the metastasis of these tumours, it was found that the majority of the diversity accumulated in the primary tumour. It is within these primary tumours where metastasis-competent subclones undergo selection [9]. The identification of subclonal mutations is clinically relevant following observations that even low-frequency clones can carry markers of prognosis and drive the process of metastasis [4]. Independent of genetic heterogeneity, Okegawa et al. suggested that intra-tumour heterogeneity also presents itself in the form of metabolic differences between tumour cells, further demonstrating the complexity present within renal cancer [10].

Current treatment strategies for RCC include antiangiogenic and immune therapies, the latter being effective in only a subset of cases [11,12,13]. Very recently, Motzer et al. performed integrative multi-omics analyses of 823 renal carcinomas from a randomised phase III clinical trial (IMotion 151) and identified seven robust molecular subtypes [14]. These molecular subgroups were associated with differential clinical outcomes following a combination of an anti-angiogenesis agent (AA; bevacizumab, anti-VEGF) and an immune checkpoint inhibitor (ICI; atezolizumab or anti-PD-L1) versus a VEGF receptor tyrosine kinase inhibitor (sunitinib).

Here, we investigate the subclonal mutation composition of clear cell renal cell carcinoma in two steps (Figure 1). In the initial discovery phase, we analysed two spatially separated biopsies and a matched normal sample from each of 16 ccRCC patients to provide an overview of the diversity and to inform the selection of genes for the second in-depth follow-up analysis. In the second phase, we used the constructed gene panel to sequence 826 genes at high coverage in 178 paired tumour samples and 89 matched normal samples from 89 ccRCC patients. We found frequent subclonal mutations in *TP53* and in mucin coding genes *MUC6*, *MUC16*, and *MUC3A.* Further, we tested for clonal exclusivity to identify combinations of signalling pathways that co-exist in the same tumour but in different tumour cell clones.

## 2. Results

The assessment of whole-exome and transcriptomic sequencing data from paired tumour biopsies along with a matched normal biopsy from 16 ccRCC patients revealed an average of 40% of mutations were private and 31% of genes were differentially expressed in only a single biopsy. A gene panel was produced consisting of 826 genes and targeted for sequencing at deep coverage in a larger cohort of 89 patients, each with paired tumour biopsies and a matched normal biopsy. We found that the mutational frequencies of the most commonly aberrated genes in ccRCC found in our cohort are comparable to those expected given data from previous large cohort studies. With the identification of low-frequency mutations following deep sequencing, the average number of private mutations increased to 62%. After the assignment of mutations to clones using the tool Cloe, enrichment and pathway-level clonal exclusivity analysis was applied to identify clonally exclusive pathway pairs.

### 2.1. Genetic and Transcriptomic Diversity in 16 Ccrcc Patients

The coverage of the whole-exome sequencing (WES) data was on average 85x, and mutation calling (see Methods) identified between 29 and 130 single-nucleotide variants (SNVs), insertions, and deletions (indels) per patient (Figure 2A). The fraction of mutations that was only detected in one of the two biopsies from the same tumour was on average 40%, which indicates high levels of intra-tumour genetic diversity. These mutations are referred to as private, whereas mutations detected in both tumour samples of a patient are known as shared. 

From the RNA-sequencing (RNA-seq) data, the differentially expressed genes were called by comparing each tumour sample to its paired normal using both single and paired-end data (see Methods). We found an average of 6364 genes per patient to be upregulated and 6598 genes downregulated (Figure 2B) with an average of 31% of differentially expressed genes being detected only in one of the two biopsies. Pathway overrepresentation analysis was performed with the set of differentially expressed genes using the Reactome pathway database [15]. Among the most overrepresented pathways are many pathways related to translation, signal transduction and growth factors (Figure 2C). The signalling pathways involving the growth factors PDGF, VEGF, SCF, or the growth factor receptor EGFR are deregulated in many patients. Of note, the vascular endothelial growth factor A (VEGFA), important for angiogenesis, cell growth, and survival [11] is upregulated in all patients of this data set. The most overrepresented pathways related to translation are highly overrepresented in patients 4, 15, and 16. They are enriched only privately in one tumour sample of patients 2, 3, 14 and 15 each, indicating that these deregulated processes are subclonal in these tumours. Assessment of the similarity between samples as measured by the Euclidean distance showed a clear separation between tumour and normal samples, as expected. Within patient 3, where a large number of differentially expressed genes were identified, and patient 15, where multiple privately enriched pathways were found, TU1 and TU2 samples are more distant in comparison to the other patients which cluster according to their sequencing method (Appendix A). 

### 2.2. In-Depth Sequencing of 826 Selected Genes in 89 ccRCC Patients

In the second phase, the cohort was extended to 89 ccRCC patients. From each patient, two spatially separated biopsies of the primary tumour and a matched normal sample were collected. This data set includes 14 patients from the WES data set (two samples were removed due to an insufficient amount of material), and 75 additional patients. Utilising our customised gene panel, we sequenced the 178 tumour and 89 normal samples at high depth. The deep coverage enables the detection of low-frequency mutations, and the larger cohort provides increased statistical power such that rare subclonal mutation patterns can be detected. The sequenced reads contain unique molecular identifiers (UMIs), which allows for the correction of potential sequencing errors. The coverage of the panel sequencing (panel-seq) data set was on average 933x, and after UMI consensus building and read filtering, it was 93x. The number of SNVs as well as indels in the panel-seq data set was on average 22 per patient (Figure 3, top panel). Pairwise comparison of the two biopsies from the same tumour revealed that on average, 62% of the mutations in a patient were private to one of the two samples with 87 of the 89 tumours (98%) containing at least one private mutation. Among the most frequently mutated genes, only 10 were mutated in more than 10% of the patients (only one in more than 50% of the patients), confirming the long-tail phenomenon commonly seen in cancer cohorts [16].

The four most commonly mutated genes in ccRCC, VHL, *PBRM1*, *SETD2* and *BAP1* [17], were also among the most frequently mutated genes in our data set (Figure 3, bottom panel). Three mucin genes: *MUC6 *(42%), *MUC16* (38%), and *MUC3A* (18%) were also frequently mutated in our cohort. The mutation frequencies of *VHL*, *SETD2*, and *BAP1* were 54%, 12%, and 13%, respectively and comparable to the frequencies found in the TCGA cohort. Somewhat lower mutation frequencies were seen in *TP53* (9%), *mTOR* (7%), and *KDM5C* (7%), which were also reported in previous ccRCC studies [6,17,18]. Interestingly, in nine patients, the *VHL* mutations resided in only one of the two tumour samples. The mutations in *VHL* are known to occur early in tumour development [19,20], which is in line with our observation that in 39 of 48 (81%) cases, the *VHL* mutations were shared between both tumour biopsies of a patient. Private mutation was seen in all but two tumours; these mutations were observed in one of the two tumour samples underlining the strong genetic heterogeneity of ccRCC.

The number of mutations was unequally distributed between the two tumour biopsies in each case with identified private mutations; this is pronounced in patients 16, 55 and 57. Of the most frequently affected genes (Figure 3), three private mutations were found in *VHL*, *PBRML1*, and *SETD2* within TU2 of patient 16, whereas 14 private mutations were identified in TU1. A private mutation was also found in each tumour biopsy affecting the same gene: *LAMA2*. In patient 55, all seven private mutations in the most frequently mutated genes occurred in TU2, while in patient 57, 13 private mutations were found in TU2. Like patient 16, two private mutations in each tumour sample of patient 57 impacted the same gene; however, in patient 57, that gene is LRP2. Only five patients had no mutations in the most frequently affected genes (≥7%). In patient 88, the gene *PBRM1* was hit by different mutations in the two tumour samples, demonstrating a pattern of convergent phenotypic evolution where a gene is affected by multiple distinct mutations across the clones in the tumour. One tumour sample, TU1, had a frameshift deletion and a missense mutation, while the other tumour sample, TU2, had a missense mutation at a different locus in *PBRM1*. Both subclonal missense mutations of *PBRM1* are predicted to be deleterious according to the SIFT annotation [21]. Among the most subclonally affected genes in our data set were the mucins *MUC6*, *MUC16*, and *MUC3A*. A clonal exclusivity test was applied to the cohort of 89 ccRCC patients on the gene level. This test pinpoints the gene pairs that are mutated in the same patients but tend to be mutated in different subclones, hence are mutually exclusive on the level of subclones. The most striking gene pair was TP53 and MUC16, which was clonally exclusive in patients 5 and 81 (Appendix A).

### 2.3. Pathway-Level Clonal Exclusivity in 16 Ccrcc Patients

To reconstruct the evolutionary history of the tumours and assign mutations to specific clones, we used Cloe [23]. The WES data from paired tumour biopsies and matched normal samples of 16 ccRCC patients enabled us to map the mutated genes to pathways and to detect pathway pairs that are affected in several patients. The two most striking clonally exclusive pathway pairs, i.e., pathways that are aberrated in different clones of the same tumour in a mutually exclusive fashion, are “major pathway of rRNA processing in the nucleolus and cytosol” (referred to as pathway 1), “O-glycosylation of TSR domain-containing proteins” (pathway 2)}, and pathway 1, “defective B3GALTL causes Peters-plus syndrome (PpS)” (pathway 3), which are clonally exclusive in both patients in which they are affected (Figure 4A,B, Appendix A), namely, patients 8 and 14 (*p* < 10^−5^). Pathway 1 belongs to the category “metabolism of RNA”, while pathway 2 falls into the class “metabolism of proteins”, and pathway 3 is a disease pathway related to diseases of glycosylation [15]. Pathway 1 was also significantly enriched among the differentially expressed genes in 13 of 16 patients of this cohort. 

Aside from inferring the mutation-to-clone assignment, the Cloe software also estimates the fractions of the clones in each sample (Figure 4C). This is important in order to interpret possible changes on the transcriptomic level in the bulk samples. Pathway 1 was mapped to clones 1 and 2 in patient 8 (Figure 4B), which together have a clonal faction of 47.5% and 30.7% in samples TU1, and TU2, respectively (Figure 4C). Pathway enrichment analysis shows that pathway 1 is also highly overrepresented in these two samples on the transcriptomic level (Figure 4D). Pathways 2 and 3 were assigned to clone 3 which is, with 16.6%, the most abundant in sample TU2 of patient 8 (Figure 4C). Pathway 2 is also enriched among the differentially expressed genes in this sample, but pathway 3 is not suggesting that the underlying mutations seem to alter the expression of pathway 2 (Figure 4D). Pertaining to patient 14, no enrichment of pathway 1 could be found in either sample (Figure 4D). Pathway 1 was assigned to clone 2, which was estimated to have a clonal fraction of only 8.6% and 0.2% in samples TU1 and TU2, respectively (Figure 4C), which may explain why there was no signal detectable in the bulk transcriptome samples for this pathway. Pathways 2 and 3, however, were assigned to clone 3 (Figure 4B), which has an estimated clonal fraction of 0.5% and 12.8% in samples TU1 and TU2, respectively. In both bulk RNA samples, pathways 2 and 3 were enriched in the second tumour sample.

## 3. Discussion

We analysed intra-tumour heterogeneity using two ccRCC patient cohorts. In the first phase, the investigation of the WES and RNA-seq data of 16 ccRCC patients revealed that intra-tumour heterogeneity is very pronounced on the genetic level, with an average of 40% of mutations found to be private. We also found that 31% of differentially expressed genes were detected in only one of the two patient tumour biopsies. In the second phase, the extended cohort of 178 tumour biopsies and the deep sequencing coverage enabled us to detect not only rare subclonal mutations, especially in *TP53* and mucin coding genes *MUC6*, *MUC16*, and *MUC3A*, but also early genomic alterations such as *PBRM1* and *VHL* with enough statistical power.

Intra-tumour heterogeneity has been reported previously by Gerlinger et al. in 2012. Following the extraction of 30 samples from four tumours, those authors observed up to 69% of somatic mutations not to be present within all samples [2]. In the same year, the analysis of 25 single cells from one ccRCC patient revealed the large extent of genetic heterogeneity between different tumour cells [24]. These studies of intra-tumour heterogeneity in ccRCC reported patterns of convergent phenotypic evolution in several genes including *VHL, BAP1*, *SETD2*, *PBRM1*, *PIK3CA*, *PTEN*, and *KDM5C* [2,4], which were also among the most frequently mutated genes in our cohort. ccRCC development is largely driven by the loss of one gene, *VHL*. Tumourigenesis typically starts with a large deletion on chromosome 3p, followed by mutational *VHL* inactivation. In addition to *VHL*, the 3p deletion also removes one copy of *PBRM1*, *BAP1*, and *SETD2*. Since *VHL* inactivation alone is insufficient [25], mutations in *PBRM1* and *BAP1* are necessary for ccRCC development. Importantly, these mutations tend to be mutually exclusive [19]. Interestingly, we have seen *VHL* mutations in only one of the two tumour samples in nine out of eighty-nine patients. This implies that these alterations would have been missed if the other tumour piece was analysed alone, with the consequence that these tumours would have been considered as *VHL* wild-type ccRCC. While VHL mutations are considered truncal, we hypothesize that in these nine samples, the mutated allele was lost in the respective subclone [26,27,28].

In addition, some of the genes were affected by multiple distinct mutations across the clones in the tumour. This is comparable to some of our previous findings in the *VHL* gene [3]. *PBRM1* mutations occur in 19% of the patients in our cohort, which is less than the frequency reported from TCGA [17,18]. We found a pattern of convergent phenotypic evolution in *PBRM1*: the gene was hit by two different deleterious missense mutations in each of the two biopsies of one patient.

We identified frequent subclonal mutations in *MUC6*, *MUC16*, and *MUC3A*, indicating that alterations of these mucin genes may also be critical in ccRCC development. In the TCGA data set, the mucins *MUC4*, and *MUC16* were also among the seven most frequently mutated genes [17], but our mutation frequencies of *MUC6*, *MUC16*, and *MUC3A* are 42%, 38%, and 18%, higher than those reported in TCGA [17]. In previous ccRCC studies, *MUC16* was reported to be among the most recurrently mutated genes [18,29]. Our analysis of the mutation distribution within the MUC16 (also termed CA125) amino acid sequence revealed that non-silent mutations seem to be clustering at the end of the sequence in the SEA domains (Appendix A) whose precise function is not known. These extracellular SEA domains can be extensively O-glycosylated and it was suggested that they can bind nucleic acids or sugars, or be released through cleavage [30]. In this context, it is of note that several RCC studies reported an association between increased levels of MUC16, poor prognosis and advanced tumour stage, suggesting the potential use of MUC16 as a serum biomarker in RCC [31,32,33]. Although *MUC16* had one of the highest mutation frequencies, our immunohistochemistry analysis showed no expression of MUC16 in the ccRCC samples (data not shown, see methods for IHC protocol). How mutations in SEA domains affect MUC16′s function and contribute to its release in the serum in ccRCC patients remains to be evaluated. It was shown that in lung cancer, *MUC16* mutations can lead to its oncogenic upregulation [34] and the overexpression of *MUC16* is associated with increased tumour cell growth, cancer cell migration, and resistance to cytotoxic drugs [35]. Recent studies also discovered frequent non-silent *MUC16* mutations in breast cancer [36,37], another cancer type in which *MUC16* was observed to be overexpressed [38]. Furthermore, *MUC16* mutations have been implicated as cancer-driving in a pan-cancer analysis that assessed the functional impact of mutations on differential gene expression profiles [39].

The majority of mucin gene mutations found in our cohort have a low variant allele frequency (VAF). Specifically, more than 75% of the mutations in *MUC6*, *MUC16*, and *MUC3A* have a VAF below 10%, and almost half below 5%. Given the mostly low VAFs of mutated mucin genes in our study, the analysis of the effect of mutations on mucin protein expression and its prognostic value would be very challenging. MUC6 and MUC16 protein expression is hardly detectable in ccRCC and MUC3A show weak to moderate expression in all ccRCC analysed (Human Protein Atlas and own data (MUC16), not shown). Despite the difficult interpretation of varying positivity of *MUC3A*, an increased expression of this mucin was correlated with poor prognosis in localised ccRCC [40]. 

Besides *MUC6*, *MUC16* and *MUC3A*, additional mucins may play an important role in ccRCC, as their expression level was shown to be predictive of clinical outcome. Decreased expression of Mucin 4 and Mucin 18 predicted poor prognosis [41,42] whereas high Mucin 7, Mucin 5A, and Mucin 13 expression was found to be associated with worse patient outcome [43,44,45].

*TP53*, a well-known tumour suppressor, was found to be mutated in less than 10% of ccRCC [17,18,46,47], which is confirmed in the cohort analysed here (9%). *TP53* mutations were associated with the reduced survival of renal cancer patients [48,49]. Of note, a previous study of the intra-tumour diversity in ten ccRCC cases revealed that mutation in *TP53* were one of the most extreme examples of gene mutations being detected more often when sequencing multiple biopsies per tumour instead of a single one [4]. We confirmed this finding as four of eight mutations were detected in only one of the two tumour samples. This observation suggests that *TP53* mutations may be a crucial subclonal event in ccRCC and explains the low prevalence of *TP53* mutations in earlier studies. Motzer et al. have evaluated somatic alterations across different histological subtypes and reported a lower prevalence of *PBRM1* mutations in ccRCC with sarcomatoid differentiation, whereas *TP53* mutations had an increased prevalence in non-ccRCC with sarcomatoid differentiation. Sarcomatoid RCC exhibited a highly proliferative phenotype with high immune presence and PD-L1 expression, explaining increased sensitivity to therapeutic intervention with atezolizumab+bevacizumab versus sunitinib [50,51]. Similar to non-ccRCC with sarcomatoid differentiation, subclonal *TP53* mutations could also be a first molecular step into the development of an aggressive phenotype of ccRCC leading to sarcomatoid differentiation. Motzer et al. have recently identified seven ccRCC subtypes with specific angiogenesis, immune, metabolic, stromal, and cell-cycle profiles showing differential clinical outcomes to VEGF blockade alone or in combination with anti-PD-L1 [14]. These molecular clades have a differential prevalence of *TP53*, *PBRM1*, *KDM5C*, and *CDKN2A/2B* alterations. Our observation of subclonal *TP53* mutations suggests that primarily tissue samples with sarcomatoid differentiation may display high levels of intra-tumour heterogeneity.

A total of 87 tumours (98%) had private mutations and were detectable in only one of two tumour samples. Given the relatively large tumour volumes of ccRCC, with pT1 and pT2 tumours having diameters of up to 7 and 11 cm, respectively, spatial heterogeneity represents a tremendous challenge to individual therapy. Underrating the mutational burden due to spatial heterogeneity of gene alterations is thus a common problem in cancer research as well as in molecular tumour diagnostics. Single cell analysis and appropriate bioinformatics tools may help to overcome this bottleneck, particularly if only single little tumour biopsies are available.

Within a subset of patients, we see a large difference in mutational counts between the two biopsies. These differences could arise due to differences in tumour purity or perhaps aberrations affecting genes needed for DNA damage repair, resulting in an accumulation of mutations [52]. We controlled for tumour purity with our requirement of at least 70% tumour cells and so explored the potential aberrations in genes involved in DNA repair obtained from MutSigDB [53]. Missense mutations were found in both *ERCC1* and *POLR2A* in TU1 of patient 16. ERCC1, together with XPF, forms a nuclease essential for nucleotide excision repair with ERCC1 required for DNA binding [54,55]. *POLR2A* encodes the largest subunit of RNA polymerase II (RNAPII). RNAPII initiates the recruitment of transcription-coupled nucleotide excision repair factors such as CSB when stalling at DNA lesions blocking translation [56]. Dysfunction in just one of these proteins may lead to an increase in the number of mutations, as seen in TU1 of patient 16. In TU2 of patient 55, we identified downstream intron variants of *XPC* that produce a protein of the same name functioning to recognise DNA damage during the global genome–nucleotide excision repair pathway [57]. Although these *XPC* variants have not been classed as altering XPC function, given the large difference seen between TU1 and TU2 in patient 55, these variants may be a cause of the accumulation of mutations observed in the second biopsy.

In an attempt to identify co-existing clones with affected pathways related to the metabolism of proteins and RNA, we applied a clonal exclusivity test to 16 ccRCC patients on the pathway level. This test allows the identification of pathways that are perturbed in different clones of the same tumour in a mutually exclusive fashion. These pathway alterations occurred in clones that evolved in parallel along different branches of the tumour phylogeny. The two most striking pathway pairs include “major pathway of rRNA processing in the nucleolus and cytosol” (pathway 1), which is clonally exclusive with “O-glycosylation of TSR domain-containing proteins” (pathway 2) and “defective B3GALTL causes Peters-plus syndrome (PpS)” (pathway 3) (Figure 4A). The pathway pairs {1, 2} and {1, 3} are affected in the two patients through a different subset of genes (Figure 4A), namely *WDR18* and *THSD4* in patient 8, and *DDX49* and *ADAMTS14* in patient 14. ADAMTS14 belongs to the ADAMTS protein family, which are secreted zinc metalloproteases that play a role in the extracellular matrix related to angiogenesis and cancer [58]. THSD4 is also referred to as ADAMTS-like protein 6 and is also secreted to the extracellular matrix [59]. Both *ADAMTS14* and *THSD4* contain the thrombospondin type 1 repeat (TSR) domain [60]. The proteins with TSR domains can undergo O-fucosylation, a protein modification that plays a role in angiogenesis and Notch signalling [60,61,62]. To conclude, the pathways detected as clonally exclusive (pathways 1, 2, and 3) are also enriched among the differentially expressed genes in some of the samples. Pathways 2 and 3 are functionally deregulated in only one of the two biopsies, showing that this deregulation is subclonal. The deregulation may arise due to the mutations in *THSD4* and *ADAMTS14*, which are members of these pathways. Whilst being of interest, the exclusivity pattern was observed in two patients, and therefore validation of these findings in a larger cohort would be beneficial.

## 4. Materials and Methods

### 4.1. Experimental Design

The analysis of intra-tumour heterogeneity and subclonal mutation patterns was comprised of two phases (Figure 1). In an initial discovery phase, whole-exome and transcriptome sequencing data from paired tumour biopsies from 16 ccRCC patients plus one matched normal sample per patient were analysed to obtain an overview of the diversity in these samples. In this first exploratory step, the detected mutations informed the design of our gene panel for the second phase. Furthermore, frequently mutated ccRCC genes from the publicly available data sets provided by the Cancer Genome Atlas Research Network (TCGA) [17] were considered for the selection of genes in the panel. During the second phase, a total of 826 selected genes were then targeted at deep coverage in an extended cohort of 89 patients for a detailed analysis of tumour heterogeneity.

### 4.2. Patient Material

Two cohorts of 16 and 89 ccRCC patients with no prior treatment were chosen for the sequence analyses. The tumours of these ccRCC patients were classified according to the 2016 WHO classification [63] and reviewed by H.M. From each patient, two tumour samples and one matched normal tissue were selected. From each frozen and FFPE tissue block, haematoxylin and eosin stained sections were prepared and reviewed by a pathologist (H.M.) to ensure tissue integrity. Only tumours with at least 70% tumour cells were included in our cohort. For whole exome and RNA sequencing of 48 (16 × 3) tissue samples, 5–10 frozen sections (10 μm) were used for DNA and RNA extraction. For in depth sequencing, 3 punches (0.6 mm diameter) were taken from 267 (89 × 3) formalin-fixed, paraffin-embedded tissue blocks. All tissue samples were anonymised. 

### 4.3. Whole Exome Sequencing

The first data set encompasses two spatially separated primary tumour biopsies and one matched normal sample from each of the sixteen clear cell renal cell carcinoma (ccRCC) patients. The whole exome was sequenced using the Illumina HiSeq 2000 system to obtain 101-bp paired-end reads. The computational pipeline to analyse the data was a customised version of the NGS-pipe framework [64] that included the following steps: adapter clipping and trimming of low-quality bases with Trimmomatic [65], alignment of the reads to the human reference genome version hg19 using bwa [66], and read processing with SAMtools [67], Picard tools [68], and bam-readcount. Reads were realigned locally around indels, and base qualities were recalibrated with the Genome Analysis Toolkit (GATK) [69]. Single-nucleotide variants (SNVs) were called using the rank-combination [70] of deepSNV [3], JointSNVMix2 [71], MuTect [72], SiNVICT [73], Strelka [74], and VarScan2 [75]. The rank-combination is a method that combines the results of different variant callers by integrating the ranked lists of variants to generate a combined ranking [70]. P-values of deepSNV were corrected for multiple testing with the R package IHW [76]. Indels were called using SiNVICT [73], Strelka [74], VarDict [77], and VarScan2 [75], and combining them with the rank-combination [70]. For copy number variant detection, the tool Sequenza [78] was employed. Mutations in copy number neutral regions were selected as input for Cloe [23] in order to reconstruct the evolutionary history of a tumour and to assign the mutations to different clones. In order to account for the uncertainty in the phylogenetic tree inference, Cloe was run 20 times with different seeds.

### 4.4. Transcriptomic Data Generation and Analysis

Paired-end and single-end RNA-sequencing was performed on the Illumina HiSeq 2000 system to generate 101-bp paired-end reads, and 51-bp single-end reads for the 48 samples from the initial 16 patients. For the computational analysis, the NGS-pipe framework was adapted [64]. Reads were clipped and trimmed using Trimmomatic [65], and alignment was performed with STAR [79]. Read counts for the genes were obtained with the program featureCounts [80]. Differential gene expression analysis was carried out using DESeq2 [81], comparing each tumour sample to its paired normal sample using both single and paired-end data (i.e., a 2 vs. 2 design). Genes with a q-value less than 0.01 were considered differentially expressed. The R package WebGestaltR [82] was applied to perform enrichment analysis using all differentially expressed genes (up- and downregulated) together. As a background gene list for the enrichment analysis, only expressed genes were included. More precisely, in each comparison of a patient’s tumour samples to the matched normal samples, the expressed genes were included in the background gene list if they had at least a count of 10 fragments across the tumour and normal samples.

### 4.5. Panel Sequencing

The second data set is a panel sequencing data set. It comprises an extended cohort of patients from which a selected set of genes were sequenced at higher depth. The selection of 826 genes was informed by the mutated genes detected in the WES data set, as well as from the frequently mutated ccRCC genes in TCGA [17] (see Appendix A for the 826 gene list and bed file). We generated panel-seq data from 89 ccRCC patients, including two spatially separated primary tumour biopsies, and a matched normal sample per patient. This data set comprises 14 of the patients from the WES data set, and 75 additional patients. The data were sequenced using the Illumina HiSeq 2500 system. The sequenced reads contain unique molecular identifiers (UMIs), and this allows for the correction of potential sequencing errors. Reads with identical UMIs, which are mapped to the same genomic position, come from the same DNA molecule, and therefore, the consensus sequence can be built, and the variants can be called with higher confidence.

The pipeline for analysing the sequencing data was again a customised version of the NGS-pipe framework [64] including the following steps: raw reads were clipped and trimmed using the tool SeqPurge [83]. Reads were aligned to the human reference genome version hg19 with the aligner bwa [66]. Reads were further processed using SAMtools [67], Picard tools [68] and bam-readcount. Local realignment around indels was carried out with GATK [69]. We used the software UMI-tools [84] to group reads with identical UMI and identical mapping position together, and an in-house tool to build the consensus sequence and thereby correct sequencing errors. Our in-house tool takes the grouped reads with identical UMIs and identical mapping positions and attempts to generate the consensus sequence from these grouped reads. If the reads contain contradicting bases at a nucleotide position, it is masked with the base “N”. The SNV and indel calling was similar as for the WES data set. Some of our samples are from formalin-fixed paraffin-embedded (FFPE) material. FFPE samples are known to harbour artificial C > T and G > A alterations [85,86]. They occur mostly at lower frequencies in the range between 1–10% variant allele frequency (VAF), since the DNA damage occurs at different genomic positions in different cells [85,87]. To remove potential FFPE artefacts, we filtered out C > T and G > A mutations that had a VAF < 10%. The tool Cloe [23] used for the tree inference requires as input mutations in copy number-neutral regions. In order to filter out mutations that are in potential copy number variant regions, mutations that are within 4000 bps of an imbalanced heterozygous germline SNP were filtered out. An imbalanced heterozygous germline SNP is a SNP that has a VAF between 40–60% in the normal sample, but in the tumour sample the VAF is out of these bounds, indicating a potential copy number change. Finally, in order to perform quality control we used Qualimap [88] and FastQC [89] in the WES and panel sequencing data sets.

### 4.6. Testing for Pathway-Level Clonal Exclusivity

The mutations detected in the WES data set were assigned to clones with Cloe [23], mapped to genes, and subsequently, the genes were mapped to pathways using the Reactome pathway database [15]. This procedure resulted in a total of 877 affected pathways.

For functional annotation of the variants, SnpSift [90] and SnpEff [22], as well as the data bases COSMIC [47] version 80, and dbSNP [91] version 138 were used. In order to identify pathways that are altered in a clonally exclusive fashion, we employed the statistical test implemented in GeneAccord [92]. We only kept genes with a potential impact for this analysis. While mutations such as synonymous or intronic variants are informative for the tree inference, they were not of interest for the clonal exclusivity test. Non-silent mutations are more likely to change the phenotype of the clones and therefore these mutations are potential candidates for inducing clonal interactions. For the estimation of background rates of clonal exclusivity and co-occurrence, it is therefore important to focus on non-silent mutations in order to have accurate estimates of their clonally exclusive background distribution. To filter out silent mutations, we used the annotation program SnpEff, which classifies variants into four categories based on the potential impact of the mutation [22]. These are, in descending order of importance, the categories “HIGH”, “MODERATE”, “LOW”, and “MODIFIER”. Examples of the category “HIGH” would be frameshift indels. Missense mutations and inframe indels are classified as “MODERATE”. The category “LOW” includes synonymous and splice region mutations. Variants that are annotated as “upstream”, “intronic”, or “UTR region” fall into the class “MODIFIER”. For the GeneAccord-based clonal exclusivity analysis, we kept mutations that are in the category “HIGH”, “MODERATE”, and from the class “LOW” we kept all variants with the exception of: synonymous variants, or mutations that are annotated as the case where a start codon mutates into another start codon, or analogous for stop codon. To sum up, we kept variants such as missense, frameshift or inframe indel or variants in splice regions, but filtered out variants that are synonymous, intronic or in the UTR regions.

### 4.7. Statistical Analysis

For the data analysis in R [93] as well as for visualising results, several R packages were used including biomaRt [94,95], caTools [96], dplyr [97], ggplot2 [98], ggpubr [99], gtools [100], maxLik [101], tibble [102], magrittr [103], reshape2 [104], RColorBrewer [105], ComplexHeatmap [106], and survival [107].

### 4.8. Immunohistochemistry

TMA sections (2.5 μm) on glass slides were subjected to immunohistochemical analysis stained using Ultra Discovery (Ventana, Roche Diagnostics, Rotkreuz, Switzerland). MUC16/CA125 was immunostained using monoclonal mouse anti-MUC16 antibody (clone X75, cat. no. M1-90039; Invitrogen, diluted 1:1000 in Bond medium). MUC16 was made visible using IHC Refine kits (Ventana). Normal and tumour tissue (cut off: >5% tumour cells) were considered MUC16-positive if tumour cells showed unequivocal weak, moderate or strong cytoplasmic and membranous expression.

## 5. Conclusions

In summary, the systematic analysis of the clone constellations as performed here in large patient cohorts will contribute towards a better understanding of the evolutionary forces beyond mutation and selection that drive tumour evolution and will help to improve treatment strategies available for those with ccRCC.

## Figures and Tables

**Figure 1 cancers-13-02163-f001:**
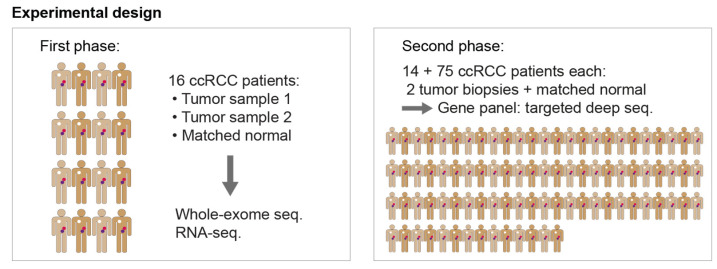
Experimental design. The first phase includes 16 clear cell renal cell carcinoma (ccRCC) patients of which two spatially separated biopsies from the primary tumour and a matched normal sample were collected. Whole-exome sequencing and transcriptome sequencing was performed and the detected mutations informed the selection of genes for the panel of the second phase. The second phase includes an extended cohort of patients and the selected genes were targeted with higher coverage. From a total of 89 patients, we analysed two spatially separated tumour biopsies and a matched normal sample per patient. Fourteen of the patients in this panel data set were also among the 16 from the first phase.

**Figure 2 cancers-13-02163-f002:**
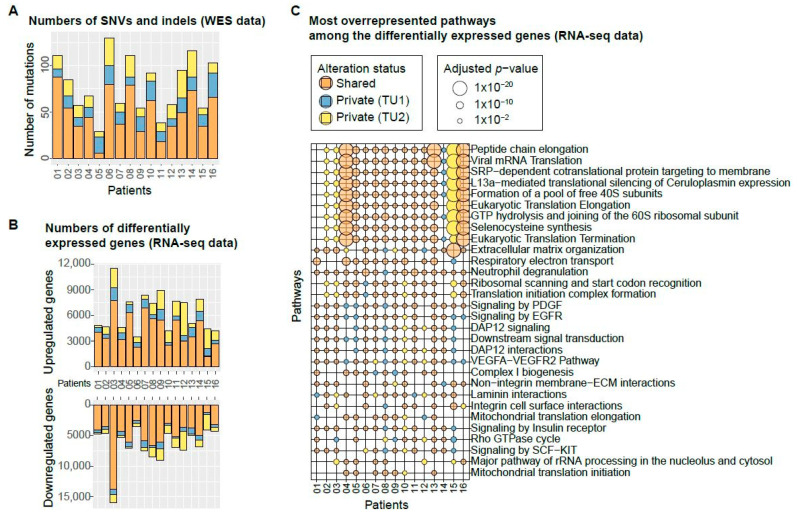
Genetic and transcriptomic diversity in 16 patients. (**A**) Number of shared (orange) and private (yellow, blue) mutations in the WES data set. A shared alteration was detected in both samples of a patient, whereas a private alteration was only found in one of the two samples. The two biopsies of the same tumour are labelled “TU1” and “TU2”. (**B**) Number of differentially expressed genes in the RNA-seq data set. (**C**) The most overrepresented Reactome pathways among the differentially expressed genes. The colour indicates the alteration status.

**Figure 3 cancers-13-02163-f003:**
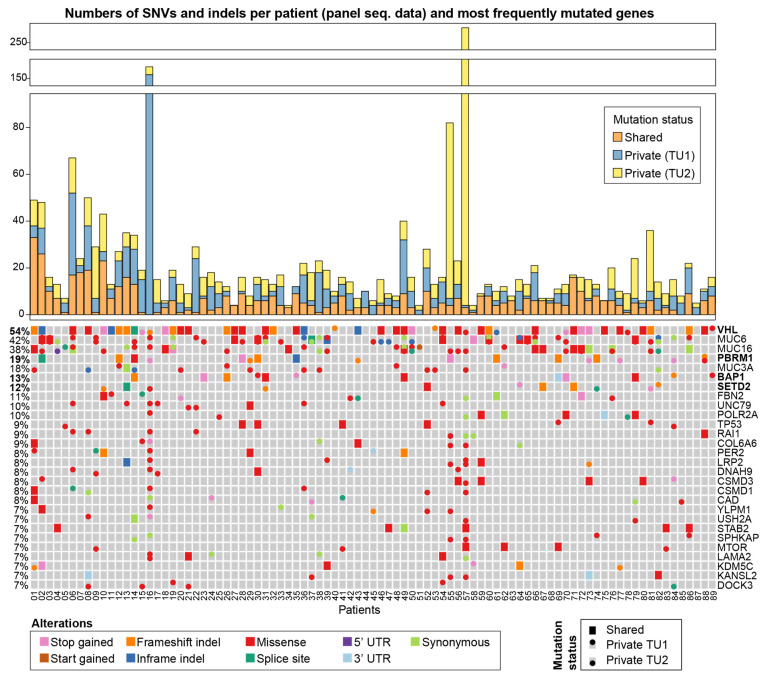
Genetic diversity in the panel seq data of 89 ccRCC patients. Top: numbers of shared and private mutations in the data set. Bottom: the heatmap highlights the mutations that were detected in the most frequently mutated genes. The four most frequently altered genes in ccRCC are *VHL*, *PBRM1*, *BAP1*, and *SETD2* [17] and are highlighted in bold. If a gene was hit by multiple mutations that all have the same status (shared, private TU1, or private TU2), the following ordering is applied to prioritise which colour is shown in the heatmap, starting with the highest priority: stop gained, start gained, frameshift indel, inframe indel, missense, splice site, five prime UTR, three prime UTR, synonymous. That means, if a gene has, e.g., a missense and a synonymous mutation, the missense mutation will be displayed in the heatmap. The variants were annotated with SnpEff [22] (Appendix A). Mutations in non-coding regions are omitted from the heatmap.

**Figure 4 cancers-13-02163-f004:**
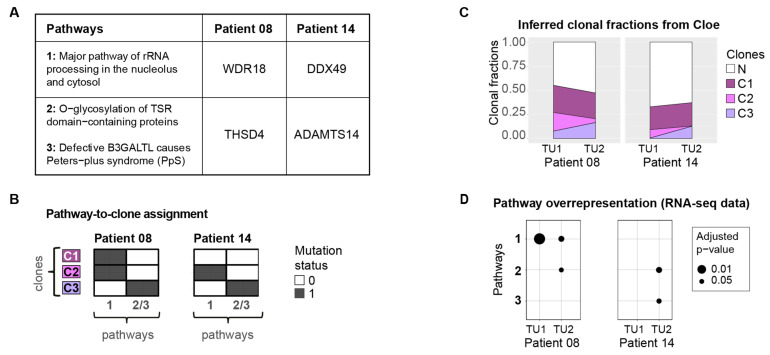
The two most striking clonally exclusive pathway pairs from the clonal exclusivity test when performed on the pathway level of the WES data set from 16 ccRCC patients. The pathway pairs are affected in patients 8 and 14 and in each patient, a different subset of genes is mutated. Hence, this clonal exclusivity pattern is only detectable on the pathway level. (**A**) The table displays the genes that are mutated in these pathways. (**B**) The heatmaps illustrate in which clones the genes in these pathways are mutated. (**C**) The proportion of cells from the clones in each of the two samples from the two patients. The label “N” represents the fraction of normal cells in the biopsy. (**D**) The data set also includes RNA-seq data from each sample. Among the differentially expressed genes in each sample, the pathways are significantly overrepresented in some of the samples.

## Data Availability

The generated read data from the clear cell renal cell carcinoma samples and the matched normal samples have been deposited in the European Nucleotide Archive under accession numbers ERP108328 and ERP108326. Full pathway-by-clone matrices from each patient have been uploaded to the Github repository (https://github.com/cbg-ethz/GeneAccord/tree/master/data/clone_pws_tbl_16_WES).

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
