# Peer review of "Spatial Distribution of Private Gene Mutations in Clear Cell Renal Cell Carcinoma"

_cancers, 2021, doi:10.3390/cancers13092163_

Round 1
Reviewer 1 Report
This work describes spatial mutational analysis performed in a large cohort of clear cell renal cell carcinomas (ccRCC, 89 patients with matched biopsies and normal tissue). The authors describe interesting observations, including frequent private mutations in one of two biopsies, specifically in TP53 and mucin coding genes. They also report on tumor gene expression analysis in 16 patients, observing that certain pathways are privately dysregulated in one of the two biopsies, suggesting subclonal differences. This work and the accompanying dataset, which will be made publicly available (although currently not accessible), will be of interest to the researchers in the field, however, a number of major concerns need to be addressed before this work is published, specifically:
- Differential expression (DE) analysis per patient:
- The DE comparison is not clearly described in results or methods, i.e. what was compared to what?
- Figure 2 legend is missing.
- Analysis is done using DEseq2, however, methods such as DEseq2 and edgeR are not designed for statistical analysis of 1 vs 1. Therefore, the authors should exercise caution when reporting such results as to not overstate the conclusions. There are a number of alternative approaches for estimating intra-tumor heterogeneity: for 2 or more samples using Euclidean distance (Sun, X. et al. Scientific reports 8.1 2018: 1-7) and per sample using tITH, sITH or DEPTH algorithms (Li, M. et al. Communications biology 3.1 2020: 1-19). The authors could try one or more of these approaches to support their findings. They could also perform DE analysis on all tumor vs normal samples (to show genes and pathways dysregulated in their cohort), and then show clustering of the key ccRCC genes per sample (again, assess the distance between paired samples)?
- On line 112, it is mentioned that VEGFA pathway is upregulated in all patients, however it is unclear if for pathway analysis all DE genes were included or just upregulated genes.
- Targeted DNA sequencing:
- List of genes included in the panel is not provided
- One of the biopsies for patients 16,55 and 57 showed significantly higher number of mutations compare to the second biopsy and to the rest of the patients. Since these samples are from formalin-fixed material, is it possible that a large number of these mutations are technical artefacts? The authors mentioned that they applied filtering of C>T/G>A variants < 10%, however it is unusual for one of the samples (spatially but not temporally different) to have much higher tumor mutation burden compared to the other, unless the tumor purity in one of the samples is suboptimal or if one of the subclones underwent a major genomic event in DNA damage repair (MMR genes, POLE etc) that enabled rapid accumulation of mutations. These possibilities should be explored in discussion, as well as it would be beneficial to add estimated tumor purity (or highest somatic VAF in CN normal regions if purity is not available) per biopsy – to help with the interpretation (were mutations called private because of the low purity?)
- Pathway-level clonal exclusivity:
- Unclear how many mutations are detected in each pathway per sample, therefore it is hard to draw any major conclusions from this analysis.
- Suggest that the authors do not place such strong emphasis on this analysis (in results, discussion and abstract), as the observation is made in only 2 patients; and it is unclear whether the pathways are indeed deregulated in different subclones, because RNAseq analysis (Fig 4D) and clonal fractions (Fig 4C – C3 observed in patient 08 in TU1 and TU2) only partially supports the findings.
- Lines 82-83, 104-105 and 212-215 – not comparing apples with apples when comparing differences on genetic and transcriptomic level. Suggest adjusting the statement to just say that heterogeneity was observed on genetic and transcriptomic level, and to not compare % of heterogeneity.
- Line 145: the sentence is missing a key part about the patient. Should start with “In patient 16”? Also, should define what TU1 and TU2 is in the main text, and on line 146 first TU2 should be TU1?
- Lines 149-150: LRP2 has private mutation/s in only 1 tumor sample for patient 16.
- Results first presented in discussion should be moved to results: lines 252,258, 269-271 (also would be good to have this presented as a supp figure), 307-308, 324-325
- Methods: rates of exclusivity are not clearly explained and unclear if patients have been pre-treated or not.
- Some of the supplementary tables/figures not mentioned in the main text: Supplementary figure S1and Supplementary table S2.
Reviewer 2 Report
This manuscript is very well written. The results are very interesting and it will add significant new knowledge to the current body of literature.
Reviewer 3 Report
The study identified substantial intra-tumor heterogeneity in ccRCC using a multi-regional sequencing strategy. This is important for understanding tumor evolution and developing future treatments. Below is a list of points to be addressed.
- It’s somewhat vulnerable to report mucin genes as cancer drivers or cancer-related genes. Mucin genes are physically large and have relatively high copy numbers, thus giving them more chances to stand out as significant genes.
- Does the second phase contain 14 patients from the first phase? What’s the reason that another 2 patients of the first phase are not included in the second phase? Please make it more clear in the manuscript.
- Figure 2(C). When you calculated the enrichment p-value, did you separate down-regulated genes from up-regulated genes? How was that reflected in your figure?
- How did you define “most subclonally affected genes”? (last sentence of section 2.2)
- It looks surprising to me that while WES gave you only 29-130 mutations for each patient, the mutual exclusivity of pathways was so significant (Table S3). Could you please provide more details about how you estimated the significance of mutual exclusivity of pathways?
Round 2
Reviewer 1 Report
The authors have sufficiently addressed the suggestions.